# The Prevailing Catalytic Role of Meteorites in Formamide Prebiotic Processes

**DOI:** 10.3390/life8010006

**Published:** 2018-02-22

**Authors:** Raffaele Saladino, Lorenzo Botta, Ernesto Di Mauro

**Affiliations:** Biological and Ecological Department, University of Tuscia, 01100 Viterbo, Italy; saladino@unitus.it (R.S.); lorenzo.botta@unitus.it (L.B.)

**Keywords:** meteorites, catalysis, formamide, prebiotic chemistry, geothermal scenarios, irradiation, radical chemistry, nucleosides

## Abstract

Meteorites are consensually considered to be involved in the origin of life on this Planet for several functions and at different levels: (i) as providers of impact energy during their passage through the atmosphere; (ii) as agents of geodynamics, intended both as starters of the Earth’s tectonics and as activators of local hydrothermal systems upon their fall; (iii) as sources of organic materials, at varying levels of limited complexity; and (iv) as catalysts. The consensus about the relevance of these functions differs. We focus on the catalytic activities of the various types of meteorites in reactions relevant for prebiotic chemistry. Formamide was selected as the chemical precursor and various sources of energy were analyzed. The results show that all the meteorites and all the different energy sources tested actively afford complex mixtures of biologically-relevant compounds, indicating the robustness of the formamide-based prebiotic chemistry involved. Although in some cases the yields of products are quite small, the diversity of the detected compounds of biochemical significance underlines the prebiotic importance of meteorite-catalyzed condensation of formamide.

## 1. The Numerous Biogenic Functions of Meteorites

Upon their arrival on the surface of this planet, meteorites deliver part of their energy into the atmosphere, with the rest being released at the impact event. The materials provided to the impact site are both a source of substrates for further reactions endowed with possible biogenic relevance, and a potential source of catalytic activity, or both. The fact that meteorites are as a whole a very complex system, prevents generalization of this list of properties and means that different meteorites are considered differently. For example, an iron-nickel meteorite can hardly be a source of organic material but it can be a source of energy and catalysis, while a chondrite could represent, in the first place, a source of catalysis and organics.

We focus here on the catalytic properties exerted by meteoric materials in the prebiotic chemistry of formamide. However, meteorite-dependent catalysis cannot be treated separately from the other functions exerted by meteorites. Hence, we firstly summarize these other functions whilst trying to sketch a general frame.

### 1.1. Meteorites Are Providers of Impact Energy during Their Passage through the Atmosphere

The first modelling of this phenomenon from a prebiotic perspective was conducted by Chyba and Sagan [1]. Part of their model has recently found theoretical and experimental support in the work of Ferus et al. [2,3,4]. Chyba and Sagan calculated that the organic syntheses driven by impact shocks were important sources of organic molecules on the early Earth, in addition to the direct delivery by extraterrestrial bodies and to the organic syntheses by energy sources such as UV or electrical discharges. The quantities of organics produced by the Heavy Bombardment (around 3.8 Gy ago) were calculated to correspond to the quantities of organics produced by other energy sources. Particularly relevant, among these organics, was HCN. On this basis, Ferus et al. have shown [4] that HCN can be produced during impact events by the reprocessing of carbonaceous and nitrogenous materials from both the impactor and the atmosphere. The investigation of high-energy events on a range of model starting mixtures shows the production of highly reactive cyano radicals and excited CO as the most abundant products initially. The subsequent reactions of these first formed excited species lead to the production of HCN and of its hydration product, formamide NH_2_CHO. In addition, it was shown [2] that the high-energy synthesis of nucleic bases from formamide can occur during the impact of an extraterrestrial body through the initial dissociation of the formamide molecule producing large amounts of CN and NH radicals. These radicals can further react with formamide to produce nucleic bases. Chyba and Sagan’s idea that the highly reactive triple-bonded HCN has a propensity to form when organic matter is challenged with sources of high energy [1], and that this energy can be provided by impactors as meteorites [1], has thus found experimental evidence [2,3,4].

### 1.2. Meteorites as Agents for Geodynamism

*Starters of Earth’s tectonics*. Plate tectonics allows chemicals essential to life to flow from the Earth’s interior to the surface. Analysis of the characteristic titanium isotopes of shales dating to 3.5 Gy has shown that more than half of the continental crust was made of felsic rocks originated from subduction [5]. Felsic rocks like granite form in the presence of heat and water in subduction events, and are indicative of plate tectonics. This datation is half a billion years earlier than expected from the temperature of the young Earth, too hot at that time to drive plate tectonics [5]. Simulation modelling [6] shows that a large single impact could trigger a subduction zone that remains active for 10 million years, providing a solution to this apparent Hadean-times contradiction. Given the known strong incidence of impacts between 4 and 3.5 Gy ago, intermittent and frequent meteorite-induced early subduction events are explained. A planet characterized by high geochemical activity was probably a more favorable scenario for life than a steady setting.

*Activators of local hydrothermal systems*. This role has been pointed out by G. Osinski and collaborators [7] and combines well with the geothermal waters scenario envisaged by A. Y. Mulkidjanian et al. [8] as a shrine for life, for which evidence has been provided [9]. Osinski first reported evidence that most impact events resulting in the formation of complex impact craters (i.e., >2–4 and >5–10 km diameter on Earth and Mars, respectively) were potentially capable of generating a hydrothermal system. The longevity and size of the hydrothermal systems can be evaluated based on the impact crater record and on the impact-generated hydrothermal deposits. These show the strong chemical activity generated at impact sites. It was suggested that hydrothermally altered and precipitated rocks can provide nutrients and habitats for a life-long period of time after hydrothermal activity has ceased. Mulkidjanian et al. presented a comprehensive discussion and data on the advantages of hydrothermal systems as sites where prebiotic phenomena could have occurred [8], essentially favored by the physical-chemical conditions characterizing these type of environments. A key property of geothermal systems is their long-term stability and the possibility of generating proficient energy cycles [9].

### 1.3. As Providers of Simple Organic Biogenic Materials, the Role of Meteorites Varies Enormously

The biogenic materials can be at first an approximation of two types: the simpler forms of atom aggregations, devoid of elaborated chemical information, like CO, CO_2_, H_2_O, NH_3_, and CH_4_, and the products of the chemical reactions occurred during the history of each meteorite starting from these simple compounds, like nucleobases, amino acids, sugars, and hydrocarbons, representing the next step of chemical complexity. Both types of materials profoundly differ among meteorites for their composition, for their relative abundances, and for the average complexity, as expected from the fact that meteorites are of the most diverse origin, composition, and history [10,11].

### 1.4. As Catalysts

The starting material selected for testing the catalytic properties of different types of meteorites was formamide. In prebiotic chemistry, formamide can be considered a substrate, a solvent, and a catalyst itself. Summarizing the reasons for the prebiotic interest of formamide:

*Simplicity*. Formamide is a central compound in the one-carbon atom chemistry domain. HCN is one of the most abundant one-carbon atom compounds in space [12], where its stability is due to the low temperature. The reaction of HCN with water leads to formamide NH_2_CHO. Formamide is also the product of the reaction between formic acid and ammonia at temperatures lower than 200–150 °C [13]. Noteworthy, formic acid was the major species observed in the Miller-Urey type experiments [14]. The formation of formamide in a Miller-Urey-type experimental set-up [3,15] and in atmosphere/meteorite impact chemistry [3] has been shown. 

Formamide is a logical starting point for reconstituting prebiotic syntheses because of its availability, stability, and solvent properties.

*Availability*. Formamide is largely available in the Universe, both far and close to us. It has been detected in interstellar space [16,17,18], in star-forming regions, and in comets [19]. The concentration of formamide in the Hadean Earth has the problem of a difficult solution, because it largely depends on the pathway of its formation, on the environment considered, and on the scenario selected; arbitrariness in this matter is high. However, the relevance of this problem greatly decreases considering the possibility of natural methods for its concentration, as thermovection [20,21], and considering the fact that geothermal volcanic scenarios are more likely [8,9] than ocean-based scenarios. The absence of oceans in the earliest times decreases the severity of the formamide-related “concentration” problem. It has been shown that formamide-based condensation reactions may occur in the presence of substantial amounts of water [22,23]. Data for the formation pathway of formamide under a space condition are reported in reference [24]. 

*Stability*. Formamide is stable as a liquid between 4 and 210 °C [24,25,26] with limited azeotropic effects. The stability in water is high (t_½_ = 199 years at 25 °C and 7.3 days at 120 under neutral pH) compared to other HCN derivatives [27].

*Properties as a solvent*. In this respect, formamide is particularly interesting as a possible prebiotic solvent alternative to water. Formamide is a polar solvent that may promote the synthesis of all types of prebiotic precursors [24,25,26], the prebiotically relevant solubilization of phosphates [28], and the phosphorylation [28,29] and the polymerization of activated monomers [30]. The earliest Earth had little or no free water [31]. Upon accumulation following its release and/or delivery, water revealed itself as mostly deleterious to unprotected biogenic precursors. The “Water Paradox” [32,33] is reasonably only solved by “Non-aqueous-Solvents first”. When it entered the scene, water most probably acted as an effector of selection, pipelining evolution based on the preferential survival of water-resistant phenotypes (as are nucleic acids polymers relative to nucleic acids monomers [34,35]) on the organization of semi-permeable membranes-contained systems.

## 2. State of the Art about the Synthetic Capacity of Formamide with Terrestrial Catalysts

A large body of previous studies has established that in the presence of catalysts of terrestrial origin, formamide affords nucleic bases (the five biologically extant ones, and others), numerous carboxylic acids, several aminoacids, condensing agents, long-chained aliphatic compounds, and a large body of miscellanea [24,25].

The catalysts analyzed were calcium carbonate, alumina, silica, zeolite (Y type), titanium dioxide, clays of the montmorillonite family, cosmic dust analogues (CDAs) of terrestrial olivines (from fayalite to forsterite), mineral phosphates, iron-sulphur and iron-copper-sulphur minerals, zirconium minerals, and borate minerals. Some of these compounds are also components of meteorites. The results showed that all the catalytic systems analyzed were active, yielding mixtures of nucleobases, aminoacids, carboxylic acids, sugars, and condensing agents, varying according to the catalyst. Results revealing the robustness of formamide chemistry are reviewed in [24,25,26]. Given that the purpose of these analyses was aimed at terrestrial scenarios, the condensation reactions of formamide with terrestrial minerals were routinely performed with thermal energy, and a variety of possible geochemical scenarios were modelled [24,25], including serpentinization-related chemical gardens [36].

## 3. The Catalytic Activity of Meteorites

Following a preliminary study on the chondrite Murchison [37], we considered the catalytic role of several types of meteorites in the chemistry of formamide using different sources of energy. The purpose of analyzing differently-powered reactions was to start exploring which scenario(s), among the plausible ones, could be fruitfully evaluated experimentally. The energies tested were: thermal [22,23,38], proton irradiation from accelerated Helium [39], and high-energy radiation from accelerated Boron [40].

The meteorites were used as powered fractions of the original sample (generally, 0–125 micro meter, with or without a pre-treatment aimed to remove eventually present endogenous organics) and selected as representative of the four major classes (iron, stony-iron, chondrites, achondrites), keeping the number of relevant sub-classes reasonably limited. Thus, comprehensive series of syntheses were performed analyzing the differently-powered catalytic activities of large panels of meteorites. Noteworthy, the meteorites performed in a very similar way with or without pre-treatments, suggesting that the mineralogical modifications eventually occurring during the pre-treatments do not significantly modify the selectivity and specificity of the formamide condensation [38].

The synthetic reactions under thermal energy were typically performed in closed vessels at 160 °C (or lower, down to 80 °C), as described in the numerous publications reviewed in [24,25]. The irradiation experiments were performed at the Accelerators of the Joint Institute for Nuclear Research (Dubna, Russia), as described in [39,40,41].

As a general conclusion, the syntheses catalyzed by meteorites in formamide are more complex and higher-yield than the corresponding ones obtained with terrestrial catalysts. However, given the complexity of the systems analyzed, exceptions are numerous and this conclusion can only be validated case-by-case (see below). Summarizing the various systems analyzed.

### 3.1. Thermal Energy-Triggered Condensations Catalyzed by Iron, Stony-Iron, Chondrites, and Achondrites

The treatment of powdered materials from 12 different types of meteorites of the four classes, preliminarily deprived of the possibly present organics, was performed at low (60 °C) and high (140 °C) temperatures [38]. Canyon-Diablo, Campo-del-Cielo, and Sikhote-Alin were the three types of Iron meteorites analyzed; Seymchan and NWA 4482 were the two Stony-Iron; NWA 2828, Gold Basin, Dhofar 959, Murchison, and NWA 1465 the five Chondrites; and NWA 5357 and Al-Haggounia the two Achondrites. References, composition, historical and terrestrial provenience, and cosmo-origin data of these meteorites are available in [38].

The results showed the one-pot synthesis of an astonishingly reach and composite panel of compounds of prebiotic interest, encompassing heterocycles (nucleic bases), carboxylic acids, amino acids, and a rich miscellanea of low-molecular-weight compounds including potential condensing agents like carbodiimide and urea. Remarkably, among the recovered nucleic bases, uracil, cytosine, adenine, guanine, and the closely related species isocytosine, dihydrouracil, and hypoxanthine, were observed. Among the amino acids we detected glycine, alanine, valine, leucine, and phenylalanine. Carboxylic acids were observed up to C-9 (nonanoic acid). In the conditions tested, the only recovered product in the absence of meteorites was purine; for a detailed analysis and discussion of the chemical aspects involved and of the meteorite-type selectivity observed, see [38]. The general order of reactivity was iron and stony-iron > achondrite and chondrite. The same condensation reactions were performed with terrestrial catalysts as the reference (troilite, pyrrothite, pyrite, chalcopyrite, volcanic basalt, olivine, hydrotalcite). The comparison showed that meteorites were more efficient catalysts than simple terrestrial catalysts, highlighting the presence of synergistic effects between the different mineralogical components of the meteorite.

The yield and the composition of the products varied as a function of the meteorite, but the panel of compounds obtained was in each case a composite and representative of a potential “prebiotic soup”. This may be taken as an indication that meteorite-catalyzed syntheses could have occurred in several of the early Earth environments for which prebiotic scenarios have been proposed, as: hydrothermal alkaline vents [42], anoxic geothermal fields [8], or less defined “warm little ponds” scenarios. In addition to the interest per se, this set of data provides a basis of comparison with the data obtained when exposing the same synthetic system to high-energy proton irradiation.

### 3.2. Proton Irradiation-Triggered Condensations Catalyzed by Iron, Stony-Iron, Chondrites, and Achondrites

A set of meteorites similar to that analyzed for thermal energy-driven reactions (including in this new study Chelyabinsk, which had fallen in the meantime) was analyzed for reactions occurring at a low temperature under irradiation with accelerated protons, thus mimicking the solar wind/flare radiation [39]. Note that a recent study suggests that solar flares could play a remarkable role in shaping the composition of the atmosphere of the primordial Earth [43]. As for thermal energy-driven reactions, the synthesis of a large panel of molecules was observed. The products isolated from the reaction mixtures were carboxylic acids as long as C20 (arachidic acid); amino acids (glycine, alanine, proline); nucleic bases (uracil, thymine, cytosine, adenine and guanine, among others); and, notably, sugars (ribose, 2′-deoxyribose, glucose, 2′-deoxyglucose, among others).

Most notably, four nucleosides, namely cytidine, uridine, adenosine, and thymidine, were synthesized. The detailed mechanism involved in the selective formation of the β-glycosidic bond has been reported for the reaction catalyzed by the chondrite NWA 1465 [41]. In summary, sugars and nucleic bases are synthesized separately from formamide by a radical cyanide (•CN) based mechanism [39], and they can then be regio- and stereo-selectively connected through the formation of a C1 sugar-centred radical intermediate [41]. Formation of the β-glycosidic bond, that so far required complex laboratory procedures and external interventions, may thus occur in a one-pot reaction.

These results show that, as an alternative to heat, radiation is a plausible and powerful energy source for prebiotic processes [39,40,41,44]. As a general trend, stony-iron, chondrite (with the only exception of Chelyabinsk), and achondrite meteorites were more active than iron meteorites. The major conclusion from this analysis is that meteorites perform as catalysts under proton irradiation conditions better than terrestrial minerals, as exemplified by the high yields obtained, and by the one-pot synthesis of molecules as complex as ribo- and 2′-deoxy ribonucleosides.

### 3.3. Heavy Ion-Triggered Condensations Catalyzed by Stony-Iron and Chondrites

The energy source of ^11^B-boron beams was studied because of the interstellar abundance of boron in our Galaxy, and because of its presence in cosmic rays near the Earth [45,46,47]. As for the reactions with Heat and Protons as energy sources, a large panel of compounds was also observed in this case, including purine and pyrimidine nucleic bases (uracil, cytosine, adenine, and guanine), nucleic base analogues, heterocycles, and carboxylic acids involved in metabolic pathways [40]. The presence of amino imidazole carbonitrile (AICN), 4,6-diamino purine (4,6-DAP), and 2,4-diamino pyrimidine (2,4-DAPy) among the observed products suggested the occurrence of unified mechanisms based on the generation of radical cyanide species (•CN). The case of differential purine versus pyrimidine nucleobase synthesis depending on the type of meteorite and of radiation used (Figure 1A–D) is noteworthy. With ^11^B beams, a lower variety of products was detected relative to heat and protons, amino acids, and nucleosides not being observed in the reaction mixtures, probably due to the different penetration depth of the ions. On the other hand, Chelyabinsk and Dhofar 959 produced higher amounts of nucleic bases, probably due to the higher linear energy transfer (LET) value, as discussed in [41]. The differential synthesis of carboxylic acids as a function of the meteorite used is shown in Figure 2. Taken together, these data suggest that the Cosmic Radiation high-energy particles can act as an efficient energy source for meteorite-catalyzed prebiotic syntheses.

### 3.4. Thermal Energy-Triggered Condensations Catalyzed by Chondrites in Formamide/Water Mixtures

General considerations [8] and geopaleontological findings [9] lead us to select a model based on geothermal fields onto whose waters, we may imagine, meteorites abundantly impacted for several hundred million years, carrying their load of endogenous and impact-related organics, providing energy and catalytic capacity locally. We have thus analyzed the catalytic effects of six different chondrites on the synthetic capacity of formamide and of formamide/water mixtures [22,23]. The chondrites were ALH 84028, EET 92042, MIL 05024, LAR 04318, GRO 95551, and GRO 95566. The waters tested were pure water, seawater, and thermal waters from the Bagnaccio volcanic area (Viterbo, Italy) and from the Phlegrean Fields (Naples, Italy). In this last instance, the meteorite NWA 4482 was analyzed. The results showed abundant syntheses, in spite of the fact that the role of water in prebiotic chemistry is controversial due to its nucleophilic character inducing solvolysis reactions and to the fact that both HCN and formamide are degraded in water. Mixtures of water and formamide afforded amino acids, nucleic bases, and carboxylic acids, as did the other systems described above. Volcanic thermal waters were the most active system for the catalytic activity of meteorites. These data hint to the important role that chondrites could have played in the early Earth subjected to the Heavy Bombardment, detailing that the mineral components that eventually reached the ground could have promoted the catalysis of organics in the hydrous and oxidizing environment that they encountered.

### 3.5. Impact-Triggered Synthesis of Nucleobases and Their Precursors in the Presence of Meteorites

Ferus et al. have studied the catalytic effect of meteorites on the formation of various nucleic bases from formamide in an extraterrestrial impact simulated with laser-zapping experiments [48]. In contrast to iron-nickel meteorites, chondritic meteorites (i.e., NWA 6472) have shown a good catalytic performance, enabling the formation of all four RNA nucleobases. When the irradiation was performed in the presence of an iron-nickel meteorite (Campo del Cielo), only cytosine, uracil, and guanine were detected among the reaction products. The lower performance of iron-nickel meteorites in the impact experiments was interpreted by the radical scavenging ability of iron. Table 1 reports the general overview of the products obtained from formamide depending on the type of energy source and meteorite.

## 4. Conclusions

The overall result of the meteorite-catalyzed condensations of formamide [22,23,37,38,39,40,41] is that in each condition tested, independent of the type of meteorite involved, the panel of compounds obtained is variegate and abundant. In addition, formamide in the presence of meteorites condenses into a plethora of prebiotic compounds, independent of the type of energy driving the reactions. A detailed analysis of the results is only possible analyzing the data reported in the specific studies on a case-by-case basis. However, the results tell us that the robustness of the formamide-meteorite chemistry is so marked that it makes it impossible to prefer one single scenario over the others. The conclusion is that looking for the components of the shrine of life, one has to look not for a single exclusive scenario, but for the one that was more prone to facilitate the next steps into complexity.

The key lesson that we learned from this ensemble of analyses is that, from a prebiotic perspective, the chemistry of formamide is very robust. From a “*chance* versus *necessity*” perspective, these first steps are thermodynamically necessary. Given the variety of energies tested, the disparate provenience of the catalysts analyzed, and the fact that the reactions may occur in pure formamide and in formamide/water mixtures (water being pure or variously mineralized), these results show that the same reactions are prone to occur in a very large set of environments. This reinforces the vision that from the point of view of prebiotic chemistry, this planet is not particularly special. The steps that follow these initial prebiotic stages, the steps leading towards complex life, are a different matter.

The key players in this logic are meteorites. Their provenience is by definition exogenous, which entails a heuristically relevant consideration: meteorites provide (in addition to minerals which are common components of this planet) catalysts which are not present on Earth. Elementally equal minerals may have very different crystal structures, explaining the observed different catalytic activities. Also relevant is that the combination of the components of the meteorites depend upon their non-terrestrial geological history.

Meteorites have formed in a number of different conditions that never occurred on Earth, enriching the catalogue of chemical possibilities beyond our terrestrial local and historically determined possibilities. This explains the interest in the observation that the large majority of products obtained through their catalysis is the same that can be obtained with terrestrial catalysts. This highlights the fact that the chemistry of HCN and of its accumulation-prone derivative formamide is likely to be met everywhere.

## Figures and Tables

**Figure 1 life-08-00006-f001:**
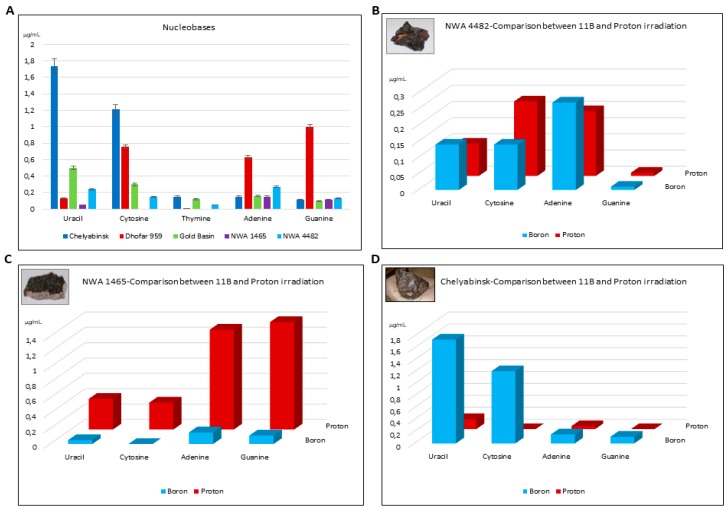
The ^11^B and Proton irradiation-powered synthesis of nucleobases from formamide in the presence of meteorites. The amount of recovered nucleobase is given as μg obtained from 1.0 mL of formamide. The reactions were carried out in the presence of the indicated meteorite (**A**). (**B**–**D**) Show that the different type of radiations yield different compounds. Data from [39,40].

**Figure 2 life-08-00006-f002:**
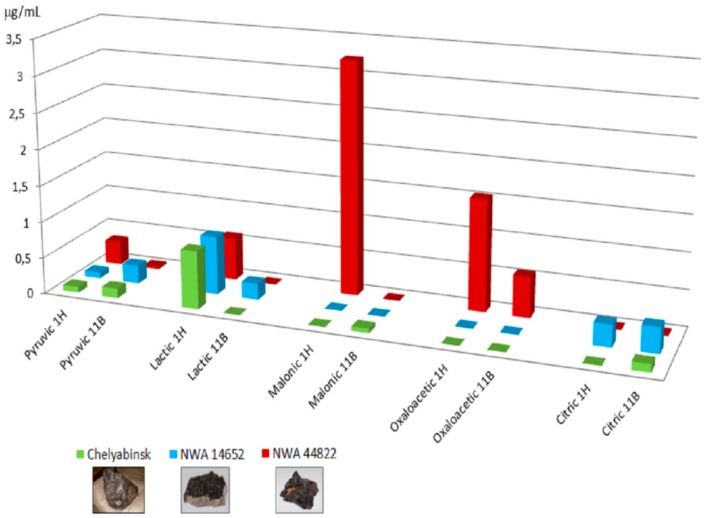
The ^11^B and Proton irradiation-powered synthesis of carboxylic acids from formamide in the presence of meteorites. Conditions as in Figure 1. Data from [39,40].

**Table 1 life-08-00006-t001:** Products obtained from formamide in the presence of different energy sources and types of meteorites.

Products	Ref.	Reaction Type	Meteorite Type
*Nucleobases and their analogs*: uracil, cytosine, adenine, guanine, isocytosine, dihydrouracil and hypoxanthine. *Aminoacids*: glycine, alanine, valine, leucine, phenylalanine. *Carboxylic acids*: from oxalic acid (C-2) up to nonanoic acid (C-9). *Condensing agents*: carbodiimide and urea.	[38]	Condensation process under thermal energy conditions in formamide	*Iron*: Canyon-Diablo, Campo-del-Cielo, and Sikhote-Alin. *Stony-Iron*: Seymchan and NWA 4482. *Chondrites*: NWA 2828, Gold Basin, Dhofar 959, Murchison, and NWA 1465 *Achondrites*: NWA 5357, Al-Haggounia.
*Nucleosides*: cytidine, uridine, adenosine, and thymidine. *Nucleobases and their analogs*: uracil, thymine, cytosine, adenine, guanine, isocytosine, hypoxanthine, 2,6-diaminopurine, and orotic acid (among others). *Sugars*: ribose, 2-deoxyribose, glucose, 2-deoxyglucose (among others). *Aminoacids*: glycine, alanine, and proline. *Carboxylic acids*: from oxalic acid (C-2) up to arachidic acid (C-20).	[39]	Formamide irradiation by high-energy proton beams	*Iron*: Canyon Diablo and Campo del Cielo. *Stony-Iron*: NWA 4482. *Chondrites*: NWA 2828, Gold Basin, Dhofar 959, NWA 1465, Chelyabinsk, and Orgueil. *Achondrites*: NWA 5357 and Al-Haggounia.
*Nucleobases and their analogs*: uracil, cytosine, adenine, guanine, isocytosine, hypoxanthine, 4,6-diamonipurine, 2,4-diamonopyrimidine, 4-amino imidazole carbonitrile, 2,4-dihydroxy pyrimidine, and orotic acid. *Carboxylic acids*: pyruvic acid, lactic acid, maleic acid, oxaloacetic acid, citric acid.	[40]	Formamide irradiation by high-energy ^11^B boron beams	*Stony-Iron*: NWA 4482. *Chondrites*: Dhofar 959, Gold Basin, Chelyabinsk, NWA 1465.
*Nucleobases and their analogs*: uracil, adenine, guanine, hypoxanthine, isocytosine, purine, 4(3H)-pyrimidinone. *Sugars*: fructose and ribose. *Aminoacids*: glycine and *N*-formyl glycine. *Carboxylic acids*: glycolic acid, oxalic acid, pyruvic acid, lactic acid, malic acid, and succinic acid.	[23]	Condensation process under thermal energy conditions in formamide/thermal water mixture	*Stony-Iron*: Seymchan and NWA4482. *Chondrites*: NWA2028. *Achondrites*: Al-Haggounia and NWA5357.
*Nucleobases and their analogs*: uracil, adenine, guanine, hypoxanthine, isocytosine, 2,6-diamino purine, purine, 4(3H)-pyrimidinone, orotic acid, and 2,4-diamino-6 hydroxypyrimidine. *Aminoacids*: glycine, *N*-formyl glycine, and alanine. *Carboxylic acids*: glycolic acid, oxalic acid, pyruvic acid, lactic acid, malic acid, succinic acid, oxaloacetic acid, fumaric acid, ketoglutaric acid, citric acid, palmitic acid, and stearic acid.	[24]	Condensation process under thermal energy conditions in formamide/sea and thermal water mixture	*Chondrites*: ALH 84028, EET 92042, MIL 05024, LAR 04318, GRO 95551, and GRO 95566.
*Nucleobases*: uracil, adenine, guanine, thymine, cytosine. *Aminoacids*: glycine.	[48]	Formamide energy impact-triggered synthesis	*Iron*: Campo del Cielo. *Chondrites:* NWA 6472.

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
