# Peer review of "The Prevailing Catalytic Role of Meteorites in Formamide Prebiotic Processes"

_life, 2018, doi:10.3390/life8010006_

Round 1

Reviewer 1 Report

The authors present a review about the role of meteorites in the origin and evolution of life on Earth and about the ability of meteorites to catalyze specific chemical reactions that are relevant to prebiological assembly of biomolecules. They present a wide variety of different modes of catalysis, including thermal temperature, irradiation, and others. These reactions usually result in a diverse product pool, suggesting that a one-pot synthesis of a large ensemble of potential biomolecule products, i.e. through messy chemistry, is a reasonable and important stepping point for further inquiries into the prebiotic synthesis of biopolymers. The authors also comment on how other aspects of meteoritic impact onto the Earth, e.g. catalysis of geodynamic phenomena, organics availability, and impact energy, tie into meteoritic catalysis and how each could have played a role in the abiotic assembly of the first biomolecules. The manuscript is generally well written (albeit with some grammatical errors that should be fixed, some of which are listed below in the minor comments section, but many others are not), the authors present their case fairly clearly, and the ideas flow in an easy-to-follow manner. I have some comments and questions and would recommend publication if the following can be addressed adequately.

Major Comments:

1.         The authors present many case studies for meteoritic catalysis, which results in a variety of products from the many different meteoritic classes, reaction types, and reaction conditions. The final list of potential synthesized molecules is quite large, and thus it would be great if the authors prepared a table which includes a list of all of the products that they mentioned in the main text of sections 2 and 3, the corresponding references, the type of reaction, and if possible, the meteorite type. I believe that this would add a lot of value to the manuscript as a way to visually, and in an organized fashion, present their case that the range of meteoritic catalysis is extremely expansive, and that these one-pot synthesis reactions resulting in a diverse set of products would have been incredibly important for the origin of life.

2.         Ostensibly, the manuscript heavily focuses on formamide chemistries resulting from meteoritic catalysis. However, formamide is not mentioned in the title nor in the first section of the introduction, and is only mentioned in passing in the abstract. For a paper that puts such a major emphasis on formamide and formamide chemistry, perhaps it would be important to make sure that upon reading the title, the abstract, and the first section of the introduction, that the reader knows specifically that formamide and formamide chemistry will be the focus of the review.

Minor Comments:

1.         In section 1.2, the paragraph headers “Starters of Earth’s Tectonics” (Line 56) and “Activators of Local Hydrothermal Systems” (Line 68) should be italicized.

2.         Section 1.3 focuses on the availability of organics provided by meteorites. The authors list a variety of small molecules, but there are no citations and listed. The authors also say that these small molecules can perform chemical reactions on the meteorite itself, resulting in larger complex molecules, yet none of these specific more complex molecules are listed by name. It would be great if the authors could include some examples of the potential complex molecules and also include citations for both this point and for the above-mentioned point.

3.         Section 1.4 is certainly interesting from a philosophical point of view, but there are no citations and not much is presented in this section. It does not need to be its own section unless there is something specific that it can add to the introduction. Perhaps it can be removed and simply mentioned briefly in another section of the introduction.

4.         The website in Line 102 should be made into a proper citation.

5.         “purport” in Line 152 should be “purpose”.

6.         The first sentence in Section 3 is a run-on and is hard to follow.

7.         “4” in Line 176 should be “four”.

8.         The sentence starting in Line 188 about the amino acids is a list, not a complete sentence.

9.         Line 215 should have an appositive statement initiated by bracketing commas encompassing the four nucleotides rather than a list initiated by a colon.

10.   In Line 255, “meteorite” should be plural.

11.   The figures presented in the manuscript are from reference 38. What I am a little confused about is whether this data or these figures have already been published (in which case the authors should include the original figures from reference 38), or are these figures newly generated from published or unpublished data related to reference 38?

Author Response

Referee 1.

Major Comments:

Q1: “ ………..it would be great if the authors prepared a table which includes a list of all of the products that they mentioned in the main text of sections 2 and 3, the corresponding references, the type of reaction, and if possible, the meteorite type”.

A1: Thanks, we agree with referee 1 about the relevance to add a table including the list of all of the products related to the type of energy and meteorite to improve the value of the manuscript. For this reason, we added at page 8 of the revised version a novel Table 1 reporting the general overview of the products obtained from formamide depending on the energy source and meteorite type.

Q2:         “For a paper that puts such a major emphasis on formamide and formamide chemistry, perhaps it would be important to make sure that upon reading the title, the abstract, and the first section of the introduction, that the reader knows specifically that formamide and formamide chemistry will be the focus of the review”.

A2: In accordance with the referee suggestions the title of the manuscript has been modified as “The Prevailing Catalytic Role of Meteorites in Formamide Prebiotic Processes”. Moreover:

A)       We modified the Abstract by the introduction of the following sentence referred to formamide: “Formamide was selected as chemical precursor and various sources of energy were analyzed” (page 1, line 15).

B)      We added the following sentence in the first section of the introduction: “We focus here on the catalytic properties exerted by meteoric materials in the prebiotic chemistry of formamide” (page 1, line 31)

Minor Comments:

Q1:         In section 1.2, the paragraph headers “Starters of Earth’s Tectonics” (Line 56) and “Activators of Local Hydrothermal Systems” (Line 68) should be italicized.

A1: OK, done.

Q2: “ It would be great if the authors could include some examples of the potential complex molecules and also include citations for both this point and for the above-mentioned point.

A2: Accordingly, one sentence “among them nucleobases, aminoacids, sugars and hydrocarbons” (page 2, line 86), and two more references were added in Section 1.3.

Q3:         Section 1.4 is certainly interesting from a philosophical point of view, but there are no citations and not much is presented in this section. It does not need to be its own section unless there is something specific that it can add to the introduction. Perhaps it can be removed and simply mentioned briefly in another section of the introduction.

A3:  OK, Section 1.4 has been removed from the text of the revised version of the manuscript.

Q4.         The website in Line 102 should be made into a proper citation.

A4.         A proper reference has been added in the revised version.

Q5: “purport” in Line 152 should be “purpose”.

A5: OK, done

Q6:  The first sentence in Section 3 is a run-on and is hard to follow.

A6: In accordance with the referee suggestion the sentence has been modified as follow: “Following a preliminary study on the chondrite Murchison [36], we considered the possible catalytic role of several type of meteorites in the chemistry of formamide using different sources of energy”.

Q7:         “4” in Line 176 should be “four”.

A7: OK, done

Q8:       The sentence starting in Line 188 about the amino acids is a list, not a complete sentence.

A8: The sentence has been modified.

Q9:         Line 215 should have an appositive statement initiated by bracketing commas encompassing the four nucleotides rather than a list initiated by a colon.

A9: OK, done

Q10:   In Line 255, “meteorite” should be plural.

A10: OK, done

Q11.   The figures presented in the manuscript are from reference 38. What I am a little confused about is whether this data or these figures have already been published (in which case the authors should include the original figures from reference 38), or are these figures newly generated from published or unpublished data related to reference 38?

A11. The figures in the manuscript are a graphic elaboration of data already published in references 38 and 39.  Note that these figures are not reported in the latter references.

Reviewer 2 Report

The present manuscript by Saladino et al. aims to summarize their work mixing meteorite material with formamide to try and find compounds of pre-biotic interest. As such, the manuscript does a reasonable job covering the extensive literature that the authors have already published on this subject. However, and critically, the manuscript does not sufficiently describe any of geological parameters necessary to justify the title of the manuscript, "The Prevailing Catalytic Role of Meteorites in Prebiotic Processes". Though this is clearly a highly interdisciplinary topic, the manuscript as written does not provide any insights into what role the mineralogy may be playing. As such, it does not add significant value to the extensive body of literature the authors have already published, and in many cases the sweeping generalizations the authors employ are likely to do more harm than good for prebiotic chemists interested in the catalytic potential of meteorites. For this reason, I cannot recommend that the manuscript be published as currently written; it requires input from an actual expert in meteorite mineralogy to provide important insights and guidance both on this manuscript and, indeed, the work upon which it relies. More specific comments are below:

In many respects, this manuscript reads more like a hypothesis paper than a review article, and it contains many controversial claims which seem out of place in a review article. For example, the notion that formamide would broaden the pre-habitability regions for life seems to ignore the observation that formamide generally requires elevated temperatures.

The structure of the review article is somewhat strange. It would be better if the authors focused on the specific areas they are knowledgeable in, rather than trying to cover areas as broad as geodynamism and panspermia (the latter of which merited only two sentences).

I found no evidence on www.astrochemistry.net to support the assertion that HCN is the most abundant "3 different atom" compound. Also, I find it hard to believe that an astronomer would make such a claim, given the known heterogeneity of “space”, between star-forming regions, molecular clouds, interstellar media, etc. Please provide literature references to support these claims.

The statements “The same condensation reactions were performed with terrestrial catalysts as reference (troylite, pyrrothite, pyrite, chalcopyrite, volcanic basalt, olivine, hydrotalcite). The comparison showed that meteorites were more efficient catalyst than their individual mineralogical components” betrays the authors collective lack of knowledge of meteoritics; implicit in this are the clear limitations of their experimental approach. First, the authors used finely powdered materials in their experiments, rather than the meteorites as they occur naturally greatly increasing the surface area. Second, the meteorites were pre-treated with base (NaOH) and acid (sulfuric), and then pyrolyzed in air at 600C; each of these steps would obviously alter the mineralogy of the meteorites. The authors have analyzed so many different, precious meteorite samples without attempting to understand how their processes are altering the material at all, nor understanding what actual mineralogical components it is made of. This could be addressed in a rather straightforward fashion by powdering the meteorite, performing XRD analysis, and then doing the clean-up process. Third, the “meteorite components” chosen by the authors are in many cases wrong, or are minor components of the meteorites. Why were multiple iron sulfides used despite being minor components, but no components such as metallic iron or nickel, particularly when the authors actually used iron meteorites? Why weren’t discrete minerals such as silicon aluminum oxides (e.g., pyroxene or plagioclase) tested? What about phyllosilicates and serpentines? These are all abundant components of different members of the meteorites types discussed in this review article. That these experiments were performed without testing major components of meteorites, and without any understanding of how the pre-treatment processes were altering the minerals renders any conclusions based on differences in meteorite composition or "group" as meaningless. 

The consistent generalization of meteorites as being iron, stony-iron, chondrite or achondrite belies the major differences in composition that are possible between these classifications. For example, Mars and lunar meteorites are achondrites, but so are brachinites. Do the authors think there are no differences between them because they're all achondrites? Similarly, CI chondrites and CB chondrites are both carbonaceous chondrites, but CIs have almost no metal, whereas CB chondrites can be up to 80 volume percent metal. Given this, statements like “As a general trend, stony-iron, chondrite (with the only exception of Chelyabinsk) and achondrite meteorites were more active than iron metorites” imply a rigor and level of detail that are entirely inconsistent with the small number of meteorite samples that have actually been analyzed and the differences between the different meteorite classifications, both gross and subtle.

In the conclusion, there is another claim that is not supported by date presented and summarized in this manuscript:

“meteorites provide (in addition to minerals which are common components of this planet) catalysts which are not present on Earth. Elementally equal minerals may have very different crystal structure, explaining the observed different catalytic activities. Also relevant is that the combination of the components of the meteorites depends upon their non-terrestrial geological history.”

Although minerals do form different polymorphs, the bigger issue with the control experiments performed in parallel with the meteoritic experiments is that they haven't been performed with those actual materials. In fact, the actual composition of those meteorite samples has not actually been determined in any of the underlying literature, so the above conclusion is not justified.

At least three of the papers cited in this review (Saladino et al PNAS 2015; Saladino et al. 2013 Chem. Eur Jour. 2013; Saladino et al 2017 Scientific Reports) claim the authors performed GC-MS analysis of organics by holding at 100 degrees C for 2 minutes, and then ramping at a rate of 10 degrees C for 60 minutes. This would bring the final temperature to 700C, which is warm enough to start melting silicate glass such as Pyrex, let alone totally destroy the columns that the authors reported using. The authors should describe the special techniques they employed in these experiments to prevent damage to their instrument and columns because it seems like this would be of great interest to the community. 

Author Response

Referee 2

Q:  For example, the notion that formamide would broaden the pre-habitability regions for life seems to ignore the observation that formamide generally requires elevated temperatures.

A: We do not agree with the referee, formamide prebiotic chemistry works very well also at low temperature as in the case of proton beam and heavy atom beam irradiations, references 39 and 40. Furthermore, some of the more interesting products from the biological point of view, as in the case of nucleosides, are produced from formamide only at low temperature.

Q: The structure of the review article is somewhat strange. It would be better if the authors focused on the specific areas they are knowledgeable in, rather than trying to cover areas as broad as geodynamism and panspermia (the latter of which merited only two sentences).

A: The section 1.4 about panspermia has been removed from the text.

Q: I found no evidence on www.astrochemistry.net to support the assertion that HCN is the most abundant "3 different atom" compound. Also, I find it hard to believe that an astronomer would make such a claim, given the known heterogeneity of “space”, between star-forming regions, molecular clouds, interstellar media, etc. Please provide literature references to support these claims.

A: The novel reference [12] has been added about the ubiquity presence of HCN in the space. Anyway, the above mentioned sentence has been removed.

Q: First, the authors used finely powdered materials in their experiments, rather than the meteorites as they occur naturally greatly increasing the surface area. Second, the meteorites were pre-treated with base (NaOH) and acid (sulfuric), and then pyrolyzed in air at 600C; each of these steps would obviously alter the mineralogy of the meteorites. The authors have analyzed so many different, precious meteorite samples without attempting to understand how their processes are altering the material at all, nor understanding what actual mineralogical components it is made of. This could be addressed in a rather straightforward fashion by powdering the meteorite, performing XRD analysis, and then doing the clean-up process.

A: The referee claims about the experimental conditions under which our experiments have been performed only in a very partial way. For example, while is true that the meteorites (power) were pre-treated with base (NaOH) and acid (sulfuric acid), and then pyrolyzed in air at 600 °C, to remove eventually endogenous organics, the referee fails to report that the experiments have been also performed (in parallel) with the untreated meteorite, leading exactly to the same results or to extremely similar results. This means that from the catalytic results point of view the mineralogical modification eventually occurring during the pre-treatment do not modify significantly the selectivity and specificity of the formamide condensation. Sentences in Section 3 have been added accordingly in order to furnish to the reader this information.

Q: Third, the “meteorite components” chosen by the authors are in many cases wrong, or are minor components of the meteorites. Why were multiple iron sulfides used despite being minor components, but no components such as metallic iron or nickel, particularly when the authors actually used iron meteorites?

A: Iron meteorites are in practice formed only by metallic iron, therefore the use of a sample of terrestrial metallic iron as a reference would make no sense. As can be clearly deduced from the cited literature, terrestrial iron sulfides had been selected as a key minerals in the Wächtershäuser prebiotic model for the origin of metabolism.

 Q: Why weren’t discrete minerals such as silicon aluminum oxides (e.g., pyroxene or plagioclase) tested? What about phyllosilicates and serpentines? These are all abundant components of different members of the meteorites types discussed in this review article. That these experiments were performed without testing major components of meteorites, and without any understanding of how the pre-treatment processes were altering the minerals renders any conclusions based on differences in meteorite composition or "group" as meaningless.

A: The panel of terrestrial minerals to be tested is in principle too broad to be analyzed and it is certainly not the purpose of this review to consider it. This point was already clarified in the text by the sentence: “….However, given the complexity of the systems analyzed, exceptions are numerous and this conclusion is only valid if examined and validated case by case (see below)”.

Q: The consistent generalization of meteorites as being iron, stony-iron, chondrite or achondrite belies the major differences in composition that are possible between these classifications. For example, Mars and lunar meteorites are achondrites, but so are brachinites. Do the authors think there are no differences between them because they're all achondrites? Similarly, CI chondrites and CB chondrites are both carbonaceous chondrites, but CIs have almost no metal, whereas CB chondrites can be up to 80 volume percent metal. Given this, statements like “As a general trend, stony-iron, chondrite (with the only exception of Chelyabinsk) and achondrite meteorites were more active than iron metorites” imply a rigor and level of detail that are entirely inconsistent with the small number of meteorite samples that have actually been analyzed and the differences between the different meteorite classifications, both gross and subtle.

A: The cosmo-origin data, mineralogical composition and sub-type of meteorites we used in our experiments is always described in the references cited in the text. The general trend we observed is that, irrespective from these details, stony-iron, chondrite and achondrite meteorites were more active than iron meteorites in the synthesis of biomolecules from formamide. This is simply an experimental evidence as clearly stated in the text.

Q: In the conclusion, there is another claim that is not supported by date presented and summarized in this manuscript: “meteorites provide (in addition to minerals which are common components of this planet) catalysts which are not present on Earth. Elementally equal minerals may have very different crystal structure, explaining the observed different catalytic activities. Also relevant is that the combination of the components of the meteorites depends upon their non-terrestrial geological history.”Although minerals do form different polymorphs, the bigger issue with the control experiments performed in parallel with the meteoritic experiments is that they haven't been performed with those actual materials. In fact, the actual composition of those meteorite samples has not actually been determined in any of the underlying literature, so the above conclusion is not justified.

A: The referee's observation is not clear. In the aforementioned sentence any consideration is made regarding the different reactivity between meteorites and terrestrial minerals. We merely note that (as is well known) in meteorites there are also amorphous and crystalline forms different from those of the corresponding terrestrial minerals, and that a different elemental composition can determine (as expected) a different catalytic activity.

Q: At least three of the papers cited in this review (Saladino et al PNAS 2015; Saladino et al. 2013 Chem. Eur Jour. 2013; Saladino et al 2017 Scientific Reports) claim the authors performed GC-MS analysis of organics by holding at 100 degrees C for 2 minutes, and then ramping at a rate of 10 degrees C for 60 minutes. This would bring the final temperature to 700C, which is warm enough to start melting silicate glass such as Pyrex, let alone totally destroy the columns that the authors reported using. The authors should describe the special techniques they employed in these experiments to prevent damage to their instrument and columns because it seems like this would be of great interest to the community.

A: In “Saladino et al 2017 Scientific Report” any GC-MS analysis has been described. In this latter case, UHPLC-MS/MS, Q Exactive-orbitrap mass analyzer, and MALDI-TOF procedures were applied. About the other two citations, any operator of the GC-MS knows very well that the temperature ramp sets in the instrument is interrupted at the holding temperature, which in the case of a standard chromatographic column corresponds to 280 °C. These are very operational details that are normally omitted in the method description, since it is referred to experts in the field. Therefore, it is not necessary any special technique to conduct the analysis, just know how to use a GC-MS.

Reviewer 3 Report

The manuscript entitled “The Prevailing Catalytic Role of Meteorites in Prebiotic Processes” by Saladino et al. reviews roles of meteorites in prebiotic chemistry. In particular, they emphasized importance of reaction products of formamide in the presence of meteorites. The authors review well recent works including their own researches. I agree that formamide has been an important starting material to produce various biologically-relevant organic compounds, leading to the emergence of life. Even though we cannot understand the detailed reaction mechanism and the role(s) of meteorites for the prebiotic chemistry, meteoritic minerals must work as catalyst to promote the reactions of formamide. One of my concerns is the definition of “meteorite” in the manuscript, where the “meteorite” is used at different level ranging from asteroidal collision to cosmic dust. The authors had better explain at first about mass, energy and composition dependent on the size of extraterrestrial materials. As the authors used a 0-125 micro meter fraction of meteorite grains (refs [35], [36]), the catalytic effect may be exerted due to the grain size. In addition, the concentration of formamide is crucial under natural environment, although the authors used pure formamide in their experiments (refs [35], [36]). The authors should provide rationale for the occurrence of formamide. Basically I have no objections to publish this paper in life after minor corrections. Specific comments are provided as follows. Line 17: biologically-relevant compounds rather than “biogenic compounds”? Line 41: Please describe the age period for “the Heavy Bombardment” Line 67: I do not understand a “geochemically” dynamic Planet. Please explain specifically about “geochemically” Line 84: I do not understand “biogenic materials”. Line 125: I do not agree with “the formamide-based pre-habitability zone relative to that of water”, if authors meant that formamide was more abundant than water in pre-habitability zone. H2O molecules are much more abundant than formamide in interstellar medium as well as in solar nebulae. Line 134: A proper reference is needed for “Earliest Earth had little or no free water”. In the absence of water, formamide cannot be produced by H2O addition of HCN. Line 149: Which are impact-induced minerals? I cannot find an impact-induced mineral in the list. Line 149&151: Please describe briefly about “The results”. Line 175: “achondrites” instead of last “chondrites” Line 193: “troilite” instead of “troylite” Line 196: Meteorite is a heterogeneous assemblage of various minerals as mentioned. The “synergistic effects” of meteoritic minerals would be discussed more. Line 286: There is no “pletora” in my English dictionary.

Author Response

Referee 3

Q: One of my concerns is the definition of “meteorite” in the manuscript, where the “meteorite” is used at different level ranging from asteroidal collision to cosmic dust. The authors had better explain at first about mass, energy and composition dependent on the size of extraterrestrial materials.

A: We thanks the referee for this suggestion. This aspect has been clarified by adding a novel sentence in Section 3 (page 4, line 161).

Q: In addition, the concentration of formamide is crucial under natural environment, although the authors used pure formamide in their experiments (refs [35], [36]). The authors should provide rationale for the occurrence of formamide.

A: A new sentence and reference 24 are reported in the revised version of the manuscript to provide rationale for the formation of formamide under space conditions (page 3, line 117).

Q: Line 17: biologically-relevant compounds rather than “biogenic compounds”?

A: OK, done

Q: Line 41: Please describe the age period for “the Heavy Bombardment”

A: OK. done

Q: Line 67: I do not understand a “geochemically” dynamic Planet. Please explain specifically about “geochemically

A: The sentence has been revised as follow: “A Planet characterized by high geochemical activity was probably a more favorable scenario for life than a steady setting

Q: Line 84: I do not understand “biogenic materials”.

A: The “biogenic materials” refers to the organic material that in principle can perform as building blocks for the origin of life.

Q: Line 125: I do not agree with “the formamide-based pre-habitability zone relative to that of water”, if authors meant that formamide was more abundant than water in pre-habitability zone. H2O molecules are much more abundant than formamide in interstellar medium as well as in solar nebulae.

A: We agree with the referee. The sentence “If formamide proves its worth as source (even though not necessarily exclusive) of bulk starting substrates for the path to prebiotic molecular complexity, this property would largely widen the formamide-based pre-habitability zone relative to that of water (line 125)” has been removed from the text.

Q: Line 134: A proper reference is needed for “Earliest Earth had little or no free water”. In the absence of water, formamide cannot be produced by H2O addition of HCN.

A: Reference [31] about the endogenous formation of water on Earliest Earth has been added in the manuscript. Formamide can be produced in a very large panel of experimental conditions and reagents (different from HCN) as reported in reference [24].

Q: Line 149: Which are impact-induced minerals? I cannot find an impact-induced mineral in the list.

A: The sentence “…..or can be considered as impact-induced minerals” has been removed from the text.

Q: Line 149&151: Please describe briefly about “The results”.

A: OK, done.

Q: Line 175: “achondrites” instead of last “chondrites”

A: OK, done.

Q: Line 193: “troilite” instead of “troylite”

A: OK, done.

Q: Line 196: Meteorite is a heterogeneous assemblage of various minerals as mentioned. The “synergistic effects” of meteoritic minerals would be discussed more.

A: OK, done.

Q: Line 286: There is no “pletora” in my English dictionary.

A: “Pletora” was modified as “plethora”.

Round 2

Reviewer 2 Report

The present manuscript summarizes results showing that meteorite powders can drive the formation of prebiotically relevant compounds from HCN. These results are not surprising considering the complex suites of organics that are actually found in many meteorites, serving as authentic evidence of their reactive nature. However, very few insights about the identities of the catalytic materials in meteorites, or mechanisms for catalysis, can be gleaned from the present review, which is still essentially completely devoid of any geochemical perspective. This research, which explores the effects of inorganic catalysts on prebiotic chemical reactions, is by definition interdisciplinary and would greatly benefit from geochemical expertise. After the first set of revisions, the manuscript still does not address the geology portion of the work in any meaningful way, and, consequently makes a number of incorrect generalizations and draws unsubstantiated conclusions. As such, it is my opinion that the paper would be of little benefit to the origins of life community and I recommend it be rejected. The specific reasons I have drawn this conclusion are described below.

The manuscript does not adequately describe a major function of meteorites, namely that they were likely the source of the water (and other volatiles relevant for the origins of life) on Earth, based on D/H ratios of ocean water and water found in meteorites (section 1.3) [e.g., Russell et al. (2017) Philosophical Transactions of the Royal Society A and references therein]. This notion is important in its own right, but also poses a paradox to the assertion in the manuscript that meteorites served as catalysts for formamide-based reactions that must be anhydrous.

With the exception of olivine, the minerals tested as terrestrial catalysts are of limited relevance to meteorites because they comprise minor fractions of the meteorites or are not present at all. Because the focus of this review is on the catalytic power of meteorites for prebiotic syntheses with formamide, this is an important limitation of the single-mineral laboratory studies that is not adequately addressed. Some examples follow. 1) Iron-nickel meteorites contain predominantly iron and nickel metal – this could easily be tested using terrestrially sourced iron and nickel to determine if the catalysis exhibited by iron-nickel meteorites is due to the iron or nickel metal, or if some other minor component is actually responsible for the catalysis. Enstatite chondrites are composed primarily of the mineral enstatite, with some ferrosilite. Determining whether either of these components, or a mixture of each, is responsible for the catalysis observed from using enstatite chondrite material would be important for understanding the catalytic potential of enstatite/ferrosilite for prebiotic chemistry. Testing the catalytic potential of phyllosilicates such as serpentine that are abundant in aqueously altered carbonaceous chondrites would similarly be an important test of whether the catalytic potential of the bulk materials in these meteorites is actually driving the observed catalysis or if the catalysis is mainly driven by minor components or mixtures. These three experiments would go a long way towards understanding the catalytic power of meteorites for accelerating formamide chemistry; since olivine was previously found to be a reasonable catalyst for formamide chemistry it is reasonable to expect that these other minerals would show activity as well. That all of these minerals are also common on Earth would obviate the need to invoke extraterrestrial catalysts for driving formamide chemistry with respect to the origins of life, if they were found to be sufficient catalysts.

The manuscript now mentions (page 4) that the pre-treatment of the meteorites prior to use as catalysts has no effect on the “selectivity and specificity of the formamide condensation.” However, additional details of the pre-treatment (base wash, acid wash, heat treatment at 600C for 1 hr) would be useful for readers to understand the what the pre-treatment entails and what effects it could have on the meteorite samples. This observation, that heating in air at 600C for 1 hour, [apparently] combusting away all insoluble organic material, has no effect on the observed catalytic activity of the meteorite powders is an important set of data. Ideally, the samples would be analyzed by X-ray diffraction prior to pre-treatment to determine the initial mineralogy, and then analyzed again after treatment to determine quantitatively how the minerals were changed by treatment with base, acid and heat under oxidizing conditions. However, even without this data, it still can place important constraints on what minerals are actually involved with the catalysis. For example, any heat-labile, acid, base or redox-sensitive materials would surely be irreversibly altered by this treatment. The observation that there was no difference in catalytic activity between un-treated and treated samples means that these heat/acid/base/redox sensitive components can all be ruled out as being responsible for the observed catalysis. This is an important conclusion that should be emphasized in the review article, not overlooked.

The petrological and mineralogical data on the meteorites analyzed should also be included in this review article. This will help readers to better understand the differences between the meteorites and provide additional context for why certain meteorites might be better catalysts than others. The acronym NWA should be defined and all of the meteorites used for the research behind this review article should be mentioned in the article. Similarly, the locale abbreviations for the Antarctic meteorites need to be explained (e.g., ALH = Allan Hills, etc.)

The statement (page 4) that “The comparison showed that meteorites were more efficient catalysts than simple terrestrial catalysts, highlighting the presence of synergistic effects between the different mineralogical components of the meteorite” is not justified by the data presented in the present manuscript and related articles. As described above, the “simple terrestrial catalysts” that were tested are for the most part irrelevant to the meteorites that were actually analyzed, and the relevant bulk materials (iron/nickel metal, enstatite, phyllosilicates etc.) actually comprising those meteorites were not actually tested. Without this basic knowledge, there is no reason to invoke more complicated explanations for the catalytic prowess of the meteorite material. This statement should be removed from page 4, as well as the the statement in the conclusions on page 7 “Also relevant is that the combination of the components of the meteorites depends upon their non-terrestrial geological history”; again, no evidence is offered that the non-terrestrial geological history of the meteorites has contributed in any way to their catalytic potential.

Page 5 – the review article should report a summary of the differences in composition (ion content, pH, salinity, etc.) of the waters used in the comparison between water, seawater and volcanic thermal waters so that readers can better understand why these water samples may have given different results in the catalytic reactions.

Page 5 – the statement “The results showed abundant syntheses, in spite of the fact that the role of water in prebiotic chemistry is controversial due to its nucleophilic character inducing solvolysis reactions and to the fact that both HCN and formamide are degraded in water” directly contradicts the earlier hypothesis that water arrived late (page 3) and served “most probably as effector of selection, pipelining evolution based on the preferential survival of water-resistant phenotypes…” These statements must either be better reconciled or the earlier statement should be removed, since it is seemingly directly refuted by empirical evidence reported in this manuscript.

In figures 1 and 2 it would be useful to include the classification of the meteorite in addition to or in place of the name. The place where the meteorites were recovered does not provide any insights as to why they would give different product distributions from HCN reactions under irradiation.

The last two paragraphs of the conclusion section (page 7) draw conclusions that are unsubstantiated by the data in the present manuscript. In particular:

“meteorites provide (in addition to minerals which are common components of this planet) catalysts which are not present on Earth. Elementally equal minerals may have very different crystal structure, explaining the observed different catalytic activities.” As mentioned previously, the predominant minerals actually present in the meteorites, such as iron and nickel metals, enstatite, and phyllosilicates, which are also common on Earth, have not been tested for their catalytic activity with formamide. Furthermore, no specific examples of minerals present in meteorites but possessing different polymorphs than their terrestrial counterparts have been identified in relation to the present work. Finally, the observation that treating these materials with acid, base, and heating under oxidizing conditions does not affect their catalytic properties suggests that exotic minerals and their associated rare polymorphs are not involved in the observed catalytic activity reported in the present manuscript. One example that illustrates this is the mineral cubanite, which is the low-temperature orthorhombic polymorph of CuFe2S3 (see Berger et al. 2014 Meteoritics & Planetary Science). Cubanite has been found in primitive chondrites such as CIs, but has been experimentally determined to undergo an irreversible transition to isocubanite (CuFe2S3 cubic polymorph) upon heating above 210 degrees C. If the isocubanite is then cooled below 210 degrees C, it breaks down to Chalcopyrite (CuFeS2) and pyrrothite (FeS). Given that the compositions of the meteorite samples actually used for this research were not determined prior to or after treatment, and that the treatment can be clearly shown to alter materials that are temperature, acid/base, or redox-sensitive, the conclusion that exotic minerals and/or polymorphs are responsible for the catalytic prowess of meteorites cannot be justified and must be removed.